# Treatment of Water Contaminated with Reactive Black-5 Dye by Carbon Nanotubes

**DOI:** 10.3390/ma13235508

**Published:** 2020-12-03

**Authors:** Pierantonio De Luca, Jànos B. Nagy

**Affiliations:** Dipartimento di Ingegneria Meccanica, Energetica e Gestionale, University of Calabria, I-87030 Arcavacata di Rende (CS), Italy; janos.bnagy1@gmail.com

**Keywords:** adsorption, carbon nanotubes, dye, reactive black-5

## Abstract

Most of the dyes used today by the textile industry are of synthetic origin. These substances, many of which are highly toxic, are in many cases not adequately filtered during the processing stages, ending up in groundwater and water courses. The aim of this work was to optimize the adsorption process of carbon nanotubes to remove an azo-dye, called Reactive Black-5, from aqueous systems. Particular systems containing carbon nanotubes and dye solutions were analyzed. Furthermore, the reversibility of the process and the presence of possible degradation phenomena by the dye molecules were investigated. For this purpose, the influence of different parameters on the adsorption process, such as the nature of the carbon nanotubes (purified and nonpurified), initial concentration of the dye, stirring speed, and contact times, were studied. The solid and liquid phases after the tests were characterized by chemical-physical techniques such as thermogravimetric analysis (TG, DTA), UV spectrophotometry, BET (Brunauer, Emmett, Teller), and TOC (total organic carbon) analysis. The data obtained showed a high adsorbing capacity of carbon nanotubes in the removal of the Reactive Black-5 dye from aqueous systems. Furthermore, the efficiency of the adsorption process was observed to be influenced by the stirring speed of the samples and the contact time, while purified and nonpurified nanotubes provided substantially the same results.

## 1. Introduction

Dyes are widely used in the textile, tanning, paper, and plastic industries. The use of dyes is characterized by large losses of the latter, caused by their high solubility that also creates an economic and environmental problem of great concern. Consequently, the removal of the dye from waste water is an important environmental problem.

Different techniques and materials can be used for the treatment of water contaminated by pollutants such as photocatalysis through the use of photocatalytic agents [1,2,3,4,5,6].

Studies on the adsorption phenomenon have shown that the latter is superior to other techniques for the treatment of waste water, being low-cost, highly efficient, simple, easy to perform, and not influenced by toxic substances [7]. There are several materials used in adsorption. Natural and synthetic zeolites are often used in adsorption to remove pollutants from water [8,9,10,11]. They are often used modified to optimize the adsorption process [12,13,14,15,16,17]. Recent studies report the removal of phenol from aqueous solutions in the presence of Cu (II) ions on synthetic NaP1 zeolite and NaP1 zeolite modified with chitosan [18].

Synthetic materials called zeotypes, such as Engelhard silicate titanium, represent another important family of microporous materials with adsorbing properties. The ETS-4 and ETS-10 phases belong to this family, which differ in the size of the pores [19,20,21,22,23,24,25,26].

Carbonaceous materials represent another class of adsorbent materials. Due to their nonpolar nature, they are able to selectively adsorb nonpolar rather than polar substances [27,28,29,30]. Organic substances, as well as natural gluconate, have been successfully used for the removal of heavy metals from aqueous systems [31].

Polymers with multilevel structures containing sub-micron pores and interconnected mesopores can be used for the adsorption of hazardous pollutants [32,33,34,35,36,37]. Functionalized organ-based materials, inorganic or mineral clays, have been shown to be potential absorbers in removing various pollutants in wastewater [38,39,40,41,42]. Biochar can be used as a potential adsorbent material. It is a product of the thermal decomposition of organic material under the limited supply of oxygen at temperatures between 350 and 700 °C. Thanks to its porosity, high specific surface area, and cation exchange capacity, it can be used as an adsorbent material in the treatment of polluted water [43,44,45]. Chitosan is a linear copolymer composed of (1–4)-linked d-glucosamine, and *N*-acetyl-d-glucosamine is a polysaccharide prepared by the *N*-deacetylation of chitin. It presents several characteristics, as nontoxicity, biodegradability, biocompatibility, bioadhesivity, and bioactivity. Different studies report chitosan and chitosan-based materials as important adsorbent materials thanks to the cationic character and the presence of reactive functional groups in polymer chains. In particular, one of the major applications is based on its ability to adsorb strongly heavy and toxic metals [46].

Carbon nanotubes (CNTs) are attracting increasing interest in research as a new adsorbent material. Carbon nanotubes can be single-walled carbon nanotubes (SWCNTs) and multiwalled carbon nanotubes (MWCNTs) [47]. They are very versatile materials and can also be used in other sectors such as fiber reinforcements [48,49]. They represent very effective materials in the treatment of water contaminated by pollutants thanks to their high adsorbing capacity [50,51,52,53,54,55,56,57,58], which is given to the CNTs by a large specific area and by hollow and stratified structures within them [59,60,61], which makes an interesting alternative for removing organic and inorganic contaminants from water.

Textile dyes significantly compromise the environment by increasing the biochemical and chemical oxygen demand (BOD and COD), compromising photosynthesis, inhibiting plant growth, entering the food chain, and promoting bioaccumulation. They can promote toxicity, mutagenicity, and carcinogenicity [62].

For this reason, the treatments currently used for the purification of textile wastewater require high efficiency but also economy.

In particular, among the different methods used, there is the bioremediation of textile dyes, through the transformation or mineralization of these contaminants by the enzymatic action of plant biomass, bacteria, extremophiles, and fungi [63,64,65,66,67,68].

Moreover, wastewater contaminated with azo dyes, mainly from the textile industry, has been the subject of research using photodegradation and sonication techniques for their treatment [69,70]. Other studies have successfully highlighted the use of biopolymers [71,72] and microgel [73]. Photo-electrochemical treatments and electrocoagulation have recently been reported [74,75]. Some studies have successfully reported the combination of multiple methods [76].

Adsorption processes are always a valid option for the remediation of water contaminated by dyes. Many studies successfully report the use of microporous materials [77,78,79,80]. Other studies have used different adsorbent materials such as sawdust, activated carbon, and carbon nanotubes [81,82,83].

In particular, previous studies report the treatment of water contaminated by the diazo dye called Reactive black-5 through different methods such as photocatalytic degradation [84], microbial decolorization [85], adsorption with activated carbon and bone carbon [86], and by the Fenton oxidation process coupled with biological treatment [87].

In this work, the removal of the azo dye called Reactive Black-5 was studied, using carbon nanotubes.

The use of carbon nanotubes as adsorbent materials has the advantage of generally being fast and cheap, as they can be easily regenerated and reused. Furthermore, they do not allow the introduction of, in the systems to be purified, new substances, which can subsequently be harmful to the environment, nor substances that can react with other molecules [88,89].

The aim was to study the influence of process parameters such as the initial concentration of the dye in the solutions to be purified, contact times, and stirring speed in order to optimize the adsorption processes. A further object was to confirm both the reversibility of the adsorption of Reactive Black-5 on the carbon nanotubes and whether, during the adsorption process, the dye molecule undergoes degradation phenomena.

## 2. Experimental

The experimental activity consisted of several consecutive phases: (a) Procurement, preparation, and preliminary characterization of raw materials; (b) execution of adsorption tests with carbon nanotubes in solutions with different concentrations of dye, for different contact times and different stirring speeds; (c) characterization of the phases after the adsorption process.

### 2.1. Materials

#### 2.1.1. Multiwalled Carbon Nanotubes (MWCNTs)

The multiwalled carbon nanotubes used in this study were the same ones that have been synthesized and characterized in our previous publications; for further information, please refer to these articles reported in the references [61,83]. Specifically, they were synthesized by the catalytic chemical vapor deposition (CCVD) method of ethylene. The catalyst was prepared utilizing NaY zeolite previously dried at ca. 130 °C for 24 h. The precursor salts –Fe(NO_3_)_3_*9H_2_O (Merck, Darmstadt, Germany, assay ≥98%) and (CH_3_COO)_2_Co*4H_2_O (Merck, Darmstadt, Germany, assay ≥98%) were dissolved separately in ultrapure water. After 30 min of sonication, the solution of iron nitrate was poured into the solution of cobalt acetate followed by a 30 min sonication. Finally, the required quantity of NaY was added to the solution in a mortar and remained there for 5 min. At the end, the Fe-Co (5 wt%) catalyst deposited on NaY was dried in an oven at 130 °C.

The reaction temperature of MWCNTs was 700 °C, the time of the reaction was 20 min, the flow rate of ethylene was 800 mL/min, the quantity of catalyst Fe-Co/NaY was 0.25 g and carrier gas N_2_ was 416 mL/min. Purification of synthesized nanotubes, to dissolve the NaY zeolite, was carried out as reported in previous studies [61]. In particular, the carbon nanotubes were immersed in a hydrofluoric acid solution (40 wt%) for three days. Subsequently, after being washed with distilled water, they were dried in an oven at 130 °C for 18 h.

The yield of MWCNTs was computed as:$$\mathrm{Yield}=\left(\frac{m_{out}-m_{in}}{m_{in}}\right)\times 100$$
where m_*in*_ is the initial mass of the catalyst and m_*out*_ is the total mass including MWCNTs. The yield is equal to 1450%, showing that this catalyst is very efficient in MWCNT production.

The TGA and DTA curves of the purified MWCNTs are shown in Figure 1.

The TGA and DTA curves of the purified MWCNTs show the oxidation of the nanotubes at 600 °C and a loss of 100%, demonstrating the efficiency of the purification.

The single carbon nanotubes have an external diameter of ca. 20 nm and the average length is equal to 20 μm. The BET surface of the nonpurified MWCNTs is equal to 108.78 m^2^/g, while that of the purified sample is 118.25 m^2^/g [61]. The isotherms of N_2_ adsorption are shown in Figure 2. Adsorption isotherms show a typical profile of porous materials, but no particular differences are appreciated for purified and nonpurified nanotubes. These curves are typical of physisorption type II in the IUPAC classification. The first part tends to saturation, while the sudden increase is significant of the formation of multimolecular layers.

#### 2.1.2. Reactive Black-5 Dye

The dye used is a commercial product, called Reactive black-5 (Sigma Aldrich, Darmstadt, Germany). It is an azo compound and it has a high molecular weight of 991.82 and the following formula: C_26_H_21_N_5_Na_4_O_19_S_6_ (Figure 3a). Figure 3b shows the UV-visible spectrum of Reactive Black 5. Three bands can be clearly identified at 307 (a), 481 (b), and 600 (c) nm.

TGA and DTA data of the Reactive Black-5 are reported in Figure 3c. The two exothermal peaks at 220 and 547 °C are due to the oxidative degradation and final oxidation of Reactive Black 5.

### 2.2. Instruments

The UV spectrophotometer (UV-3100PC Shimadzu, Kyoto, Japan) was used to measure the residual concentrations of dye after adsorption tests. All experiments were performed in triplicate and the results were reported as mean values.

Differential thermal analysis (DTA) and thermogravimetric analysis (TG) (Shimadzu-60, Kyoto, Japan) were performed with an air flow of 50 mL/min, applying a heating rate of 10 °C/min.

The N2 adsorption isotherms of the samples were performed by Micromeritics ASAP 2010 (B.E.T-Unterschleißheim, Germany). All the samples, before being analyzed, were pretreated under vacuum at 200 °C for 12 h. The total organic carbon (TOC) and total inorganic carbon (TIC) content of the product solutions were measured by a TC analyzer (Shimadzu, Kyoto, Japan).

### 2.3. Preparation of Reactive Black 5 Solutions

Seven different solutions have been prepared with the following concentrations in distilled water (Table 1).

### 2.4. Adsorption of Reactive Black-5 on MWCNTs

Here, 0.03 g of MWCNTs was added into 10 mL of Reactive Black-5 solutions with different concentrations in a stirrer. The speed was 350 and 500 rpm. Finally, the MWCNTs with the adsorbed Black-5 product were filtered, dried at 125 °C for 24 h, and analyzed by TG and DTA. The filtrate was analyzed using UV-visible absorption spectroscopy.

## 3. Results and Discussion

### 3.1. Adsorption of Reactive Black 5 on MWCNTs

The Reactive Black 5 was adsorbed on both purified and nonpurified MWCNTs. Figure 4 shows, as an example, the change in color of a solution with an average concentration among those used, i.e., equal to 37 mg/L, depending on the type of nanotubes used, purified and nonpurified, and the stirring speed.

The color of the solutions changed from dark blue to transparent as a function of time of treatment. This first qualitative analysis was followed by a quantitative one, analyzing the concentration of Reactive Black 5 in the solution as a function of time. Figure 5 shows the change in concentration starting with 15 mg/L of Reactive Black 5 as a function of stirring speed.

After 30 min, the concentration dropped drastically and after 60 min, nearly 100% of the colorant was eliminated from solution. The same value was obtained for the purified and nonpurified MWCNTs and for the different times of stirring. Figure 6 shows similar changes in concentrations starting from 37 mg/L Reactive Black 5 and 45 mg/L, respectively.

The following Figure 7a–c show the percentage of dye reduction in the three distinct solutions at different concentrations. It is possible to see that after 60 min for all the systems considered, there are reduction percentages close to 100%. The differences in reduction percentages between the different systems analyzed occur mainly in the first 30 min, confirming the above statements.

The following Figure 8 shows the adsorption capacities, expressed as mg of removed dye/grams of nanotubes, in the different solutions.

For the three solutions, adsorption capacity values of around 4.5, 12, and 15 mg g^−1^ were reached for the solutions at concentrations of 15, 37, and 45 mg/L respectively, in correspondence with the abatement close to 100%. Although, as can be guessed, the saturation of carbon nanotubes has not yet been reached in the systems analyzed, these adsorption capacity values are already sufficiently important when compared with other materials used such as iron oxide nanoparticles (IONPs) [90], pumice, and walnut wood activated carbon [91].

### 3.2. Kinetic Study

For each system studied, the reaction rates were calculated, at room temperature, in different time intervals. The following relationship Rate = [C_2_−C_1_]/[t_2_−t_1_] was used, where C_1_ and C_2_ represent the concentration of the dye at the corresponding times t_1_ and t_2_. The time intervals considered were thirty minutes each and exactly in the following time ranges: [0′–30′]; [30′–60′]; [60′–90′]; [90′–120′].

The data shown in Figure 9 show that by increasing the concentration of the initial solutions up to 37 mg/L, there is a tendential increase in the reaction rate.

The highest reaction rate values generally occurred in the first thirty minutes with a tendency to decrease with time.

The variations in the kinetics of the system were mainly observed with the variation in the stirring speed, 350 and 500 rpm, rather than the type of nanotubes used (purified and nonpurified). To highlight this aspect, the following Figure 10 focuses on the reaction rate in the first 30 min as a function of the concentration of the initial solutions.

The data show that in systems treated with a stirring speed of 350 rpm, regardless of the type of nanotube, there was a quasi-linear increase in speed, which increased with the concentration of the dye solution.

A different behavior occurred for the systems treated with a stirring speed of 500 rpm, where there was an increase in the reaction speed up to a concentration of 37 mg/L after which it began to decrease.

The nonlinear behavior of the latter can be attributed to a faster saturation of the surface due to the concurrence of both the highest stirring speed and the highest concentration.

It can therefore be said that a higher stirring speed is favorable for the removal of the dye in slightly concentrated solutions. Systems with more concentrated dye concentrations require a slower stirring speed to avoid rapid flooding of the nanotube surface.

### 3.3. Characterization of MWCNTs Containing Adsorbed Reactive Black 5

A study of the thermal characteristics of the samples was carried out, to confirm the adsorption of the dye and concomitantly to investigate both variations in weight loss and interactions between carbon nanotubes-dye molecules as a function of the process parameters. Thermogravimetric analysis was carried out on the solid sample recovered after the adsorption process. In particular, thermogravimetric analyses were carried out on two sets of purified nanotubes after post-treatment. In the first case, 0.03 g of purified MWCNTs was kept under stirring for 120 min in three solutions having different concentrations, i.e., 15, 37, and 45 mg/L, in order to compare the dye behavior adsorption. Figure 11 compares the weight loss trends of the three different samples.

The three TG curves are very similar and the first weight loss occurs at ca. 320 °C. This peak is attributed to the decomposition of Reactive Black 5 as the combustion of the MWCNTs takes place at 600 °C. The temperature of 320 °C is higher than that observed for the decomposition of pure Reactive Black 5 (Figure 4), 220 °C. This is due to the great interaction between the MWCNTs and the adsorbed molecules. The second peak of decomposition at 600 °C coincides with the combustion of MWCNTs themselves. The various weight losses of the two peaks are shown in Table 2.

The DTA curves for the three samples are reported in Figure 12 and the corresponding temperatures can be found in Table 3.

Figure 12b compares the first exothermal peak for three samples. We can see that with increasing concentration of Reactive Black 5, the DTA peak extended more and more, suggesting that the combustion occurred progressively in the colorant layers. For the highest concentration of 45 mg/L, this peak extended from 290 °C to 370 °C. Thermal analysis was also carried out in order to compare the samples as a function of the various times of treatment. The sample containing 0.03 g of purified MWCNTs and 10 mL of a solution of 37 mg/L of Reactive Black 5 was thoroughly studied, and the TG curves are shown in Figure 13, while Table 4 gathers the percent losses for the two peaks.

The relative % of the first peak increased with the time of treatment, arrived at a maximum after 60 min of treatment, and decreased later on. The increase was due to the higher quantity of adsorbed Reactive Black 5, and while the maximum adsorption capacity was reached, a desorption could occur as a function of time. The DTA curves are reported in Figure 14 and the corresponding temperatures in Table 5.

### 3.4. Determination of the Carbon Content Before and After Adsorption of the Dye

Determination of the organic and inorganic carbon content inside the solutions after the adsorption process was carried out to verify any degradation phenomena of the dye. In particular, the total organic carbon, the inorganic carbon, and the total carbon content were determined on the liquid phase. The inorganic carbon stems from carbonates, bicarbonates, and dissolved CO_2_. Table 6 shows the data obtained during the treatment of MWCNTs with a solution of Reactive 5 of 37 mg/L

The initial total carbon concentration is 8 mg/L. Its value decreased with the time of treatment. The concentration of inorganic carbon increased with the time of treatment, suggesting a partial decomposition of the colorant. The TOC decreased gradually due to the adsorption of the colorant on MWCNTs.

### 3.5. Characterization of the Liquid Phase After Desorption

A further study was carried out to verify the reversibility of the process and to seek further evidence of degradation phenomena of the dye. In particular, a solution of Reactive Black-5 was prepared in DMSO (dimethyl sulfoxide) 4 mg/L. This is the reference solution. On the other hand, an aqueous solution of Reactive Black 5 of 4 mg/L was prepared, 0.03 g of purified MWCNTs was added, and the system was stirred for 180 min at 350 rpm. After separating the solid and the liquid by filtration, the so-obtained MWCNTs containing Reactive Black 5 was put in the presence of 10 mL DMSO, and the system was stirred for 120 min at 350 rpm. The UV-visible spectra of the two samples are shown in Figure 15. It can be seen that the colorant was present in the desorption solution, but its concentration was lower than that of the reference solution, showing that the partial decomposition of the colorant occurred as it was shown during the analyses of TOC.

## 4. Conclusions

The results obtained allowed us to draw the following conclusions:

The use of MWCNTs for the removal of the Reactive-Black-5 dye has proven to be highly effective. Adsorption can even reach 100%.

Adsorption tests on both purified and nonpurified MWCNTs have been carried out to verify that the presence of catalytic and zeolitic particles in the nonpurified MWCNTs has no value in the dye adsorption process. It has been shown that the catalyst, used for the synthesis of nanotubes, has no particular influence on the dye removal process. The parameters, on the other hand, such as stirring speed and adsorption times, are important. The MWCNTs with the adsorbed colorant were examined using TGA and DTA techniques. In each case, the presence of adsorbed colorant was confirmed.

Another important aspect to underline is the possible recycling of post-treatment nanotubes: As seen from the thermal analyses, the combustion temperature of the dye is lower than that of the nanotubes, so exposing the post-treatment nanotubes to temperatures of around 500 °C, it is possible to destroy the adsorbed molecules without damaging the nanotubes, which can be subsequently reused.

The adsorbed Reactive Black 5 could be partially desorbed using DMSO. However, some of the colorant is destroyed, as it was shown by analyzing the TOC and using UV-visible.

## Figures and Tables

**Figure 1 materials-13-05508-f001:**
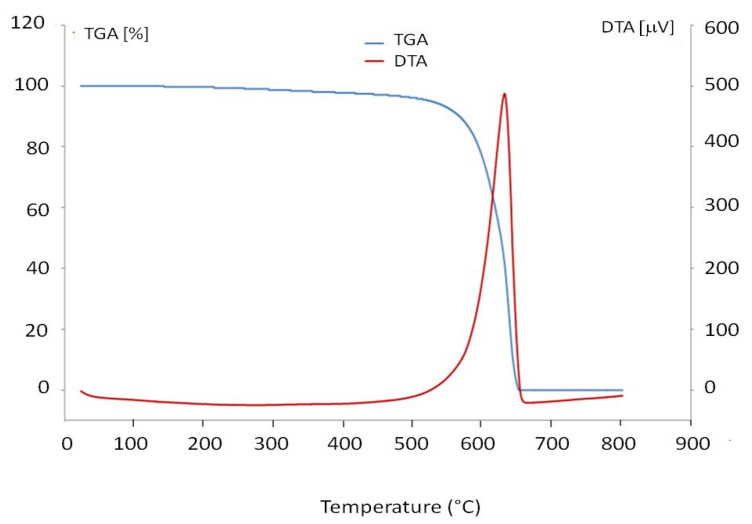
TGA and DTA curves of the purified multiwalled carbon nanotubes (MWCNTs).

**Figure 2 materials-13-05508-f002:**
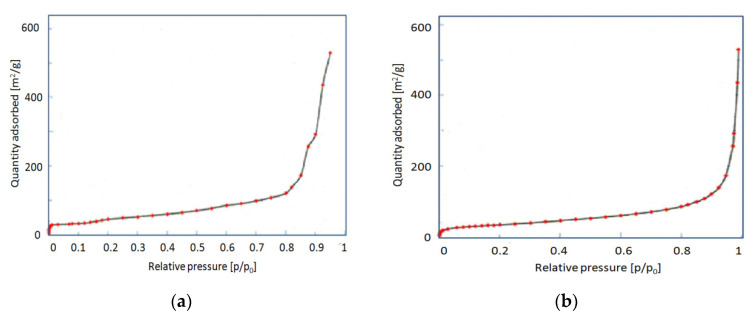
Isotherms of N_2_ adsorption: (**a**) Nonpurified nanotubes; (**b**) purified nanotubes.

**Figure 3 materials-13-05508-f003:**
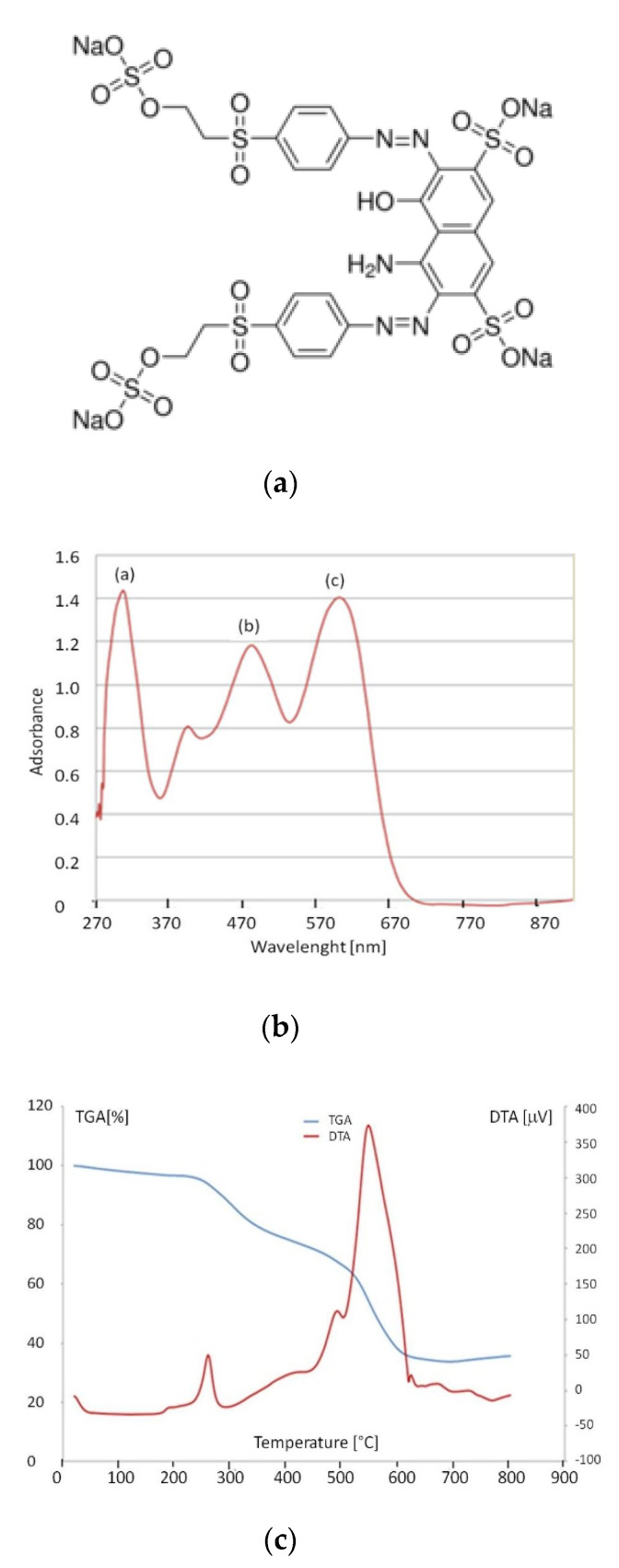
(**a**) Structural formula; (**b**) UV-visible spectrum; (**c**) TGA and DTA curves of Reactive Black-5 dye.

**Figure 4 materials-13-05508-f004:**
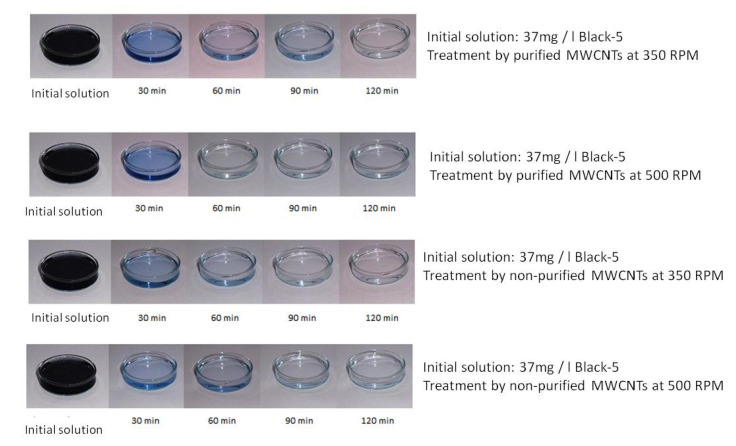
Qualitative trend of the different shades of coloring of the solution according to type of nanotubes used, purified and nonpurified, and the stirring speed (RPM-revolutions per minute).

**Figure 5 materials-13-05508-f005:**
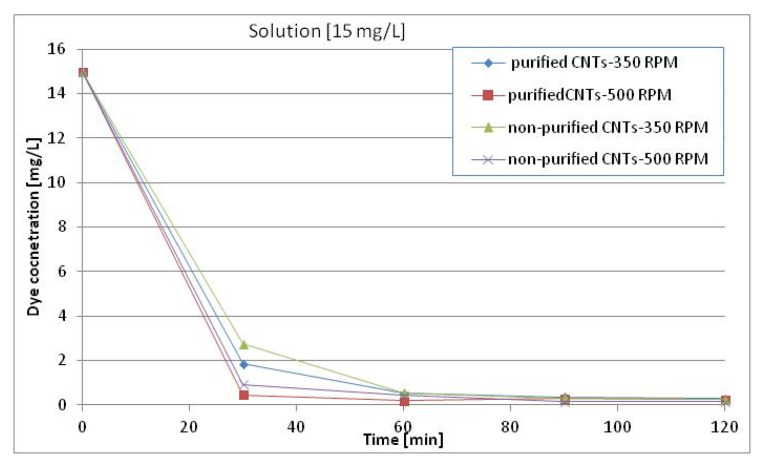
Changes in concentrations of Reactive Black-5 starting from 15 mg/L as a function of type of MWCNTs, purified and nonpurified, and stirring speed (revolutions per minute-RPM).

**Figure 6 materials-13-05508-f006:**
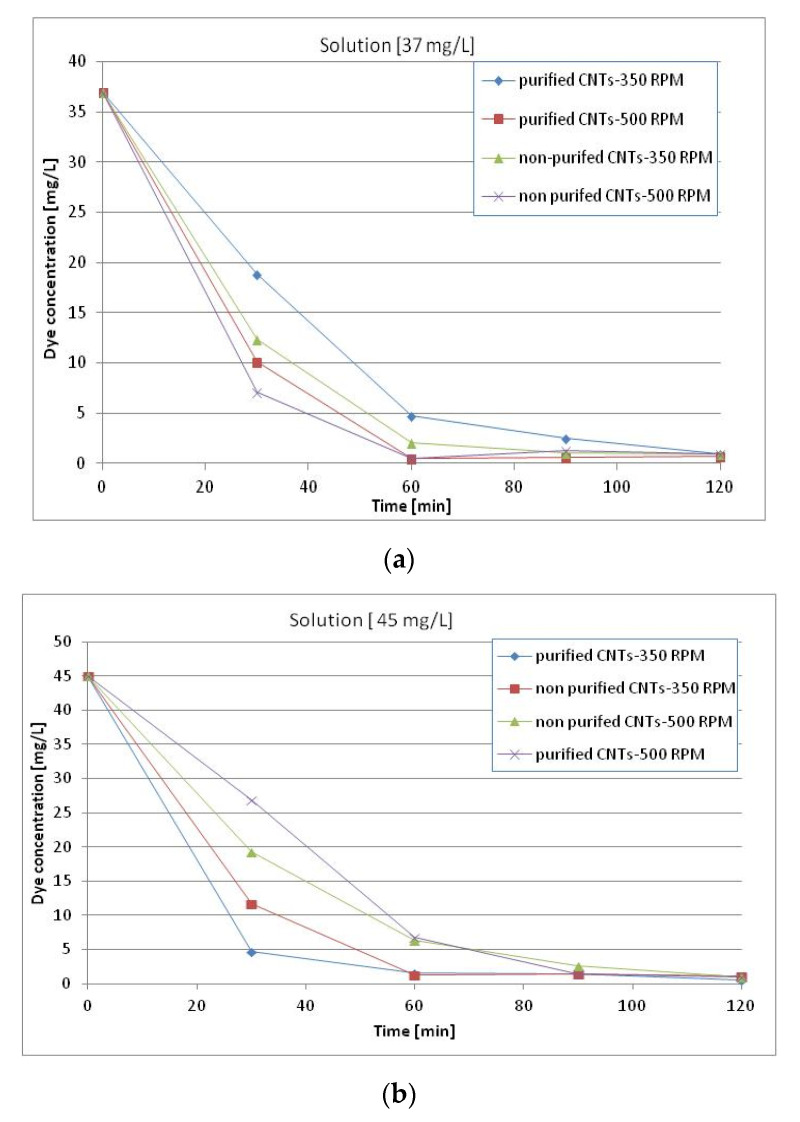
Changes in concentrations of Reactive Black-5 starting from (**a**) 37 and (**b**) 45 mg/L, as a function of type of MWCNTs, purified and nonpurified, and stirring speed (revolutions per minute-RPM).

**Figure 7 materials-13-05508-f007:**
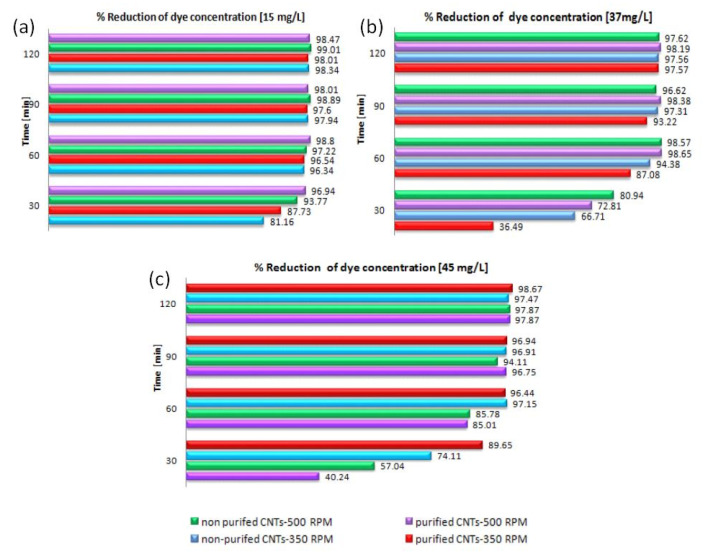
Percentage of abatement in dye solutions, depending on the type of nanotubes and the stirring speed, with initial concentrations: (**a**) 15 mg/L; (**b**) 37 mg/L; (**c**) 45 mg/L.

**Figure 8 materials-13-05508-f008:**
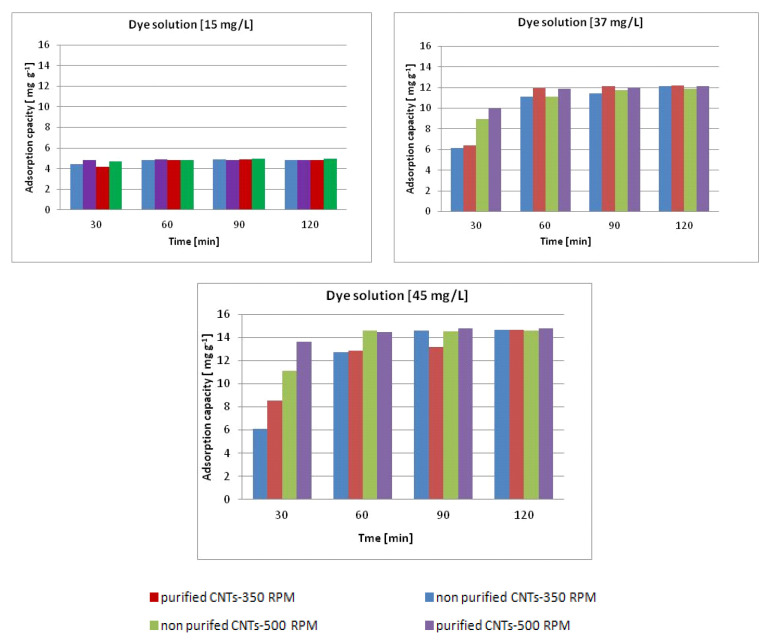
Adsorption capacity in solutions with different concentrations as a function of contact time.

**Figure 9 materials-13-05508-f009:**
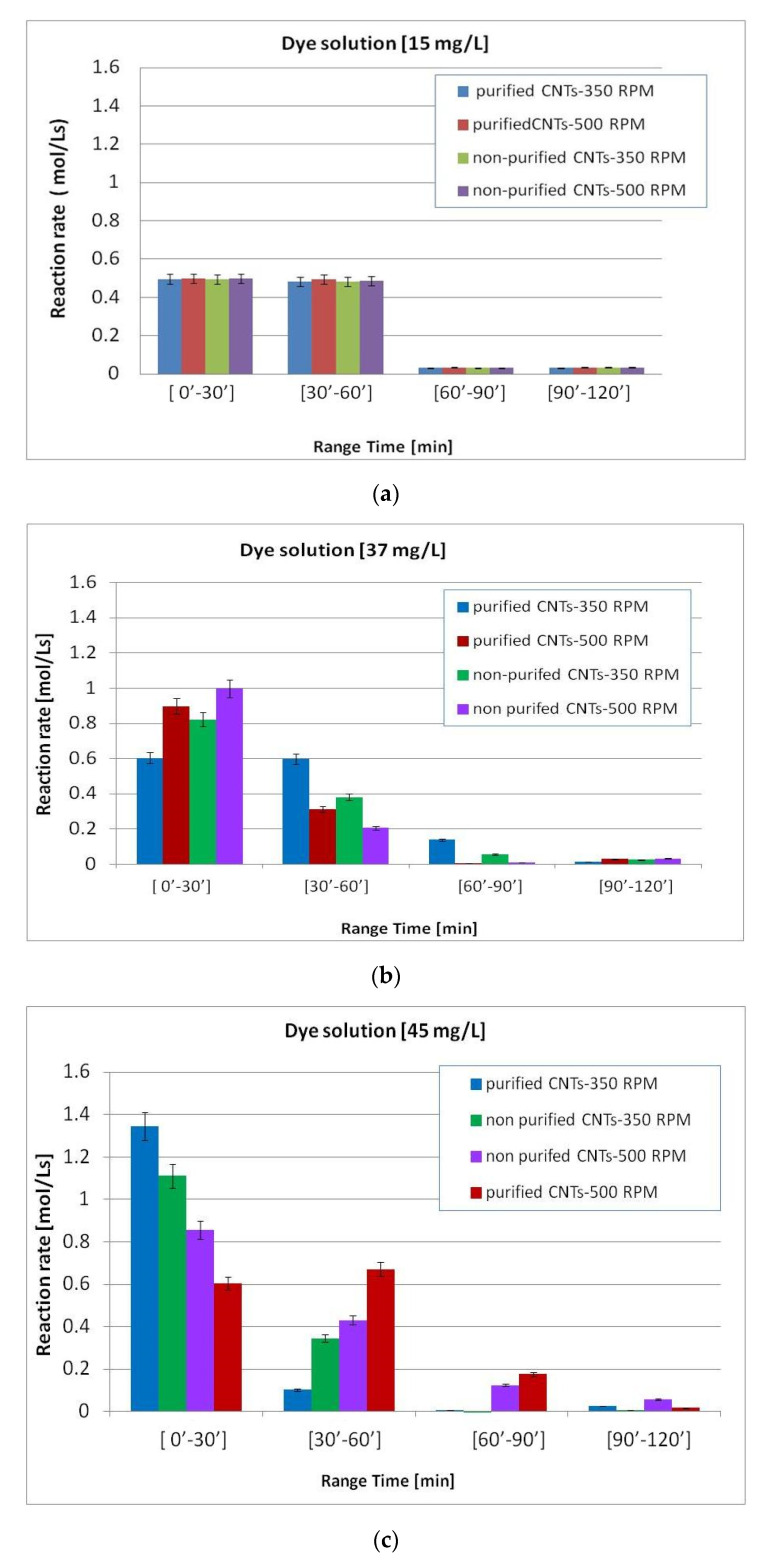
Reaction rate at room temperature at different time intervals in dye solutions with initial concentrations: (**a**) 15 mg/L; (**b**) 37 mg/L; (**c**) 45 mg/L.

**Figure 10 materials-13-05508-f010:**
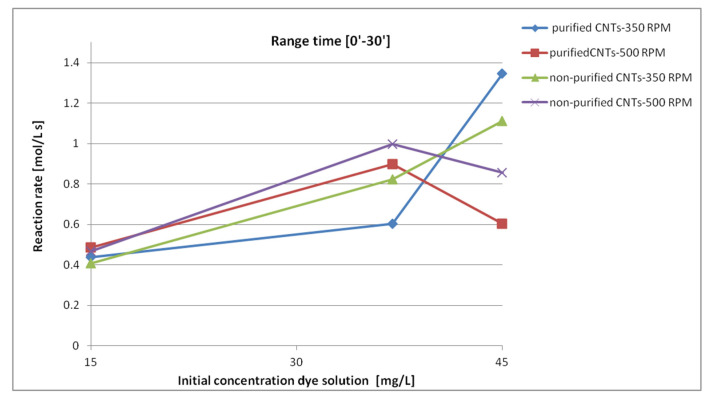
Reaction rates in the first thirty minutes as a function of the different concentrations of the dye solutions.

**Figure 11 materials-13-05508-f011:**
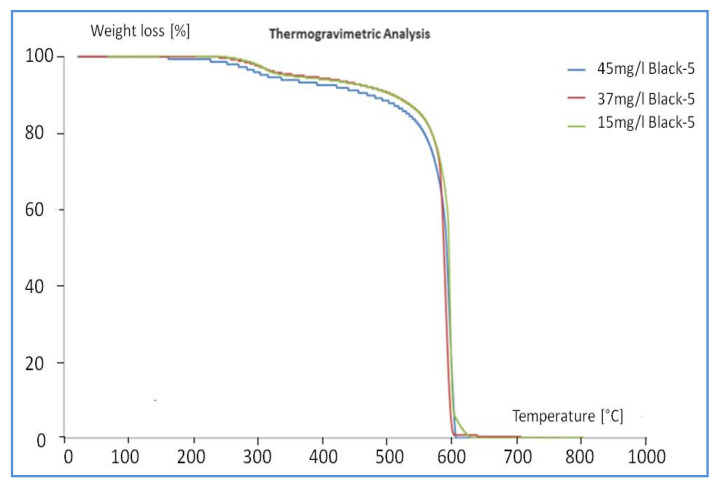
TG curves of purified MWCNTs after adsorption of the different solutions.

**Figure 12 materials-13-05508-f012:**
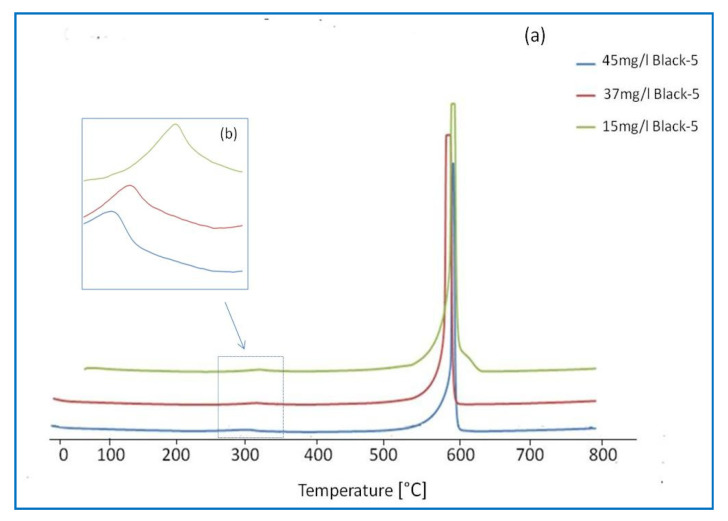
(**a**) DTA curves of purified MWCNTs after adsorption of the different solutions; (**b**) first exothermal DTA peaks.

**Figure 13 materials-13-05508-f013:**
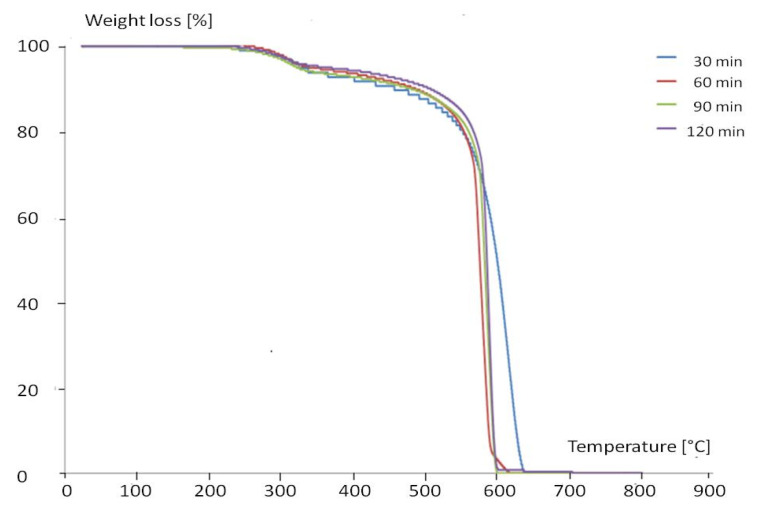
TG curves of purified MWCNTs after adsorption of 10 mL of solution of 37 mg/L of Reactive Black 5 at different treatment times.

**Figure 14 materials-13-05508-f014:**
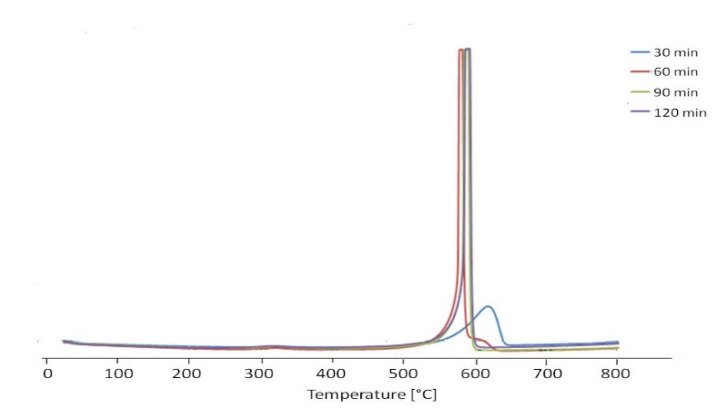
DTA curves of purified MWCNTs after adsorption of 10 mL of solution of 37 mg/L of Reactive Black 5 at different treatment times.

**Figure 15 materials-13-05508-f015:**
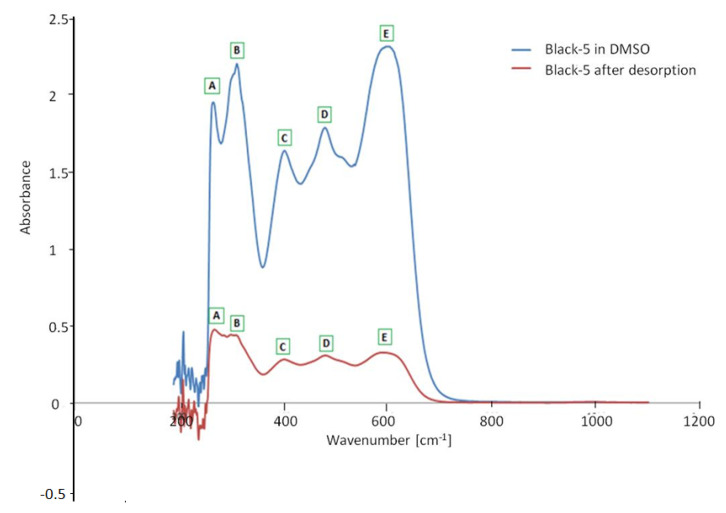
Comparison between the spectrum of Black-5 in DMSO and the post-desorption liquid phase; note how the characteristic bands of the dye (**A**–**E**) are also present in the post-desorption solution, even if with less intensity.

**Table 1 materials-13-05508-t001:** Concentrations of Reactive Black -5 solutions.

Solution	C_R Black-5_ (mg/L)
S_1_	5
S_2_	15
S_3_	30
S_4_	37
S_5_	40
S_6_	45
S_7_	50

**Table 2 materials-13-05508-t002:** Partial and total post-treatment weight losses of purified MWCNTs for different concentrations.

Concentration [mg/L Black-5]	I Loss [%]	II Loss [%]	Total Loss [%]
15	7.48	92.52	100
37	7.60	92.40	100
45	8.33	91.67	100

**Table 3 materials-13-05508-t003:** Exothermic DTA peaks of purified MWCNTs for different concentrations.

Concentration [mg/L Black-5]	I Peak [°C–exo]	II Peak [°C–exo]
15	327	596
37	323	584
45	318	591

**Table 4 materials-13-05508-t004:** Weight losses of purified MWCNTs after adsorption of 10 mL of solution of 37 mg/L of Reactive Black 5 at different treatment times.

Time [min]	I Loss [%]	II Loss [%]	Total Loss [%]
30	7.71	92.29	100
60	8.27	91.73	100
90	8.00	92.00	100
120	6.43	93.57	100

**Table 5 materials-13-05508-t005:** Exothermic DTA peaks of purified MWCNTs after adsorption of solution of 37 mg/L of Reactive Black 5 at different treatment times.

Temperature [°C]	I Peak [°C–exo]	II Peak [°C–exo]
30	327	620
60	322	582
90	323	591
120	315	593

**Table 6 materials-13-05508-t006:** Carbon content before and after adsorption of liquid phase.

Time [min]	Total Carbon [mg/L]	Inorganic Carbon [mg/L]	Organic Carbon [mg/L]
0	7.909	0.58	7.329
30	5.52	1.268	3.969
60	4.923	1.552	2.416
90	3.335	2.507	1.758
120	3.026	2.713	0.622

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
