# Peer review of "Treatment of Water Contaminated with Reactive Black-5 Dye by Carbon Nanotubes"

_materials, 2020, doi:10.3390/ma13235508_

Round 1

Reviewer 1 Report

The article is entitled: „Treatment by carbon nanotubes of water contaminated with reactive black-5 dye" but I suggest to reorganize it to give it a better sound:

"Treatment of water contaminated with reactive black-5 dye by carbon nanotubes".

The article presents simple experiments of reactive black-5 dye removal from aqueous phase using model solutions and carbon nantobues as sn adsorbent.

The abstract should be reorganized and improved according to the Journal standard which suggests: 1) Background: Place the question addressed in a broad context and highlight the purpose of the study; 2) Methods: Describe briefly the main methods or treatments applied. Include any relevant preregistration numbers, and species and strains of any animals used. 3) Results: Summarize the article's main findings; and 4) Conclusion: Indicate the main conclusions or interpretations. The abstract should be an objective representation of the article: it must not contain results which are not presented and substantiated in the main text and should not exaggerate the main conclusions.

The introduction section is too genral and short. For example: „Different are the materials that are used in adsorption such as microporous materials [9-14].” Authors should improve this paragraph and list the microporous materials used as adsorbents since they mentioned 6 articles. However, 4 of them are self-citations which consider similar material ETS-4, ETS-10. Just one reference considers natural zeolites and one - adsorbents in the removal of oil spills. Please include other materials used as adsorbents in wastewater treatment in general: like natural and synthetic zeolites, carbonaceous adsorbents, organo-minerals, chitosan and chitosan-based materials, polymers, biochars, and others. For example zeolites in Phenol Removal in the Presence of Cu(II) Ions—Comparison of Sorption Properties after Chitosan Modification, or adsorption of BTX from aqueous solutions by Na-P1 zeolite obtained from fly ash.

Also a literature review considering dyes removal from waters should be extended since many papers have been published in this subject. Please include a kind of the material, and the sorption results in the literature review.  

Authors should find a gap for their research and explain the aim of this study with emphasizing the novelty. Well-defined aim (based on the current state-of-the-art) and novelty is missing in the article.

The research consider the application of purified and non-purified carbon nanotubes. No procedures considering the purification are presented. Authors present N2 adsorption isotherms of both materials without any comment on it.

The paragraph „Instruments” constitutes just a list of used apparatus. Authors should give more details about the procedures used for the materials characterization using this instruments and justify their use. Besides, some characteristic is presented in the previous section, which is not consistent.

Based on the obtained adsorption data, the adsorption capacities of carbon nanotubes should be determined, not only the percentage removal. This will allow to compare the CNT with other materials used in dyes removal. Adsorption isotherms and kinetic studies should be also included. I suggest to provide more experimental points in the first 30 min of adsorption. This will allow to analyse kinetics of the sorption since the process of dye adsorption takes place mainly within the first 30 min.

Authors did not provide a discussion of the obtained results in relation to results obtained by other researchers. Please improve the discussion.

Please correct the caption of Figure 6. Figure 8 is not consistent. Please use the same colour for one material type, and the same order of material appearance in the figure.

In line 159 authors write that here is a linear increase in speed which increases as the concentration of the dye solution increases. However the figure 10 does not proove it.

The usage of TG/DTA method in order to characterize carbon nanotubes after dye adsorption is questionable because it does not provide valuable information. What was the point of doing this study?

It should be explained in the materials and methods section. What was the FTIR spectra of the carbon nanotubes? It will help to analyse the results. Spectra presented in figures 15 and 16 are difficult to compare because the x axis is not consistent in them and main peaks are not marked.

What is the added value of determining different kind of carbon in the liquid samples (TOC)? No discussion of the obtained results is provided. The same question considers the results presented in the figure 17.

Maybe it is not clear for the reader because the article has not well-defined aim of the study.

What was the aim of doing this research set?

Please limit self-citations, in the whole text more than 30% are self-citations.

Author Response

Dear Reviewer,

thank you for your time spent reviewing our manuscript and for giving us important tips for improving it. The manuscript has been revised in many parts. The changes have been highlighted in yellow. Here are the answers to your suggestions point by point. We hope that in this form the manuscript will find your approval to be considered for its publication.

We thank you and send you our best regards.

Answers:

(1) The article is entitled: „Treatment by carbon nanotubes of water contaminated with reactive black-5 dye" but I suggest to reorganize it to give it a better sound: "Treatment of water contaminated with reactive black-5 dye by carbon nanotubes". The article presents simple experiments of reactive black-5 dye removal from aqueous phase using model solutions and carbon nantobues as sn adsorbent.

(1)Answer. The title has been changed.

(2)The abstract should be reorganized and improved according to the Journal standard which suggests: 1) Background: Place the question addressed in a broad context and highlight the purpose of the study; 2) Methods: Describe briefly the main methods or treatments applied. Include any relevant preregistration numbers, and species and strains of any animals used. 3) Results: Summarize the article's main findings; and 4) Conclusion: Indicate the main conclusions or interpretations. The abstract should be an objective representation of the article: it must not contain results which are not presented and substantiated in the main text and should not exaggerate the main conclusions.

(2) Answer. The abstract was reorganized.

(3)The introduction section is too genral and short. For example: “Different are the materials that are used in adsorption such as microporous materials [9-14].” Authors should improve this paragraph and list the microporous materials used as adsorbents since they mentioned 6 articles. However, 4 of them are self-citations which consider similar material ETS-4, ETS-10. Just one reference considers natural zeolites and one - adsorbents in the removal of oil spills. Please include other materials used as adsorbents in wastewater treatment in general: like natural and synthetic zeolites, carbonaceous adsorbents, organo-minerals, chitosan and chitosan-based materials, polymers, biochars, and others. For example zeolites in Phenol Removal in the Presence of Cu(II) Ions—Comparison of Sorption Properties after Chitosan Modification, or adsorption of BTX from aqueous solutions by Na-P1 zeolite obtained from fly ash. Also a literature review considering dyes removal from waters should be extended since many papers have been published in this subject. Please include a kind of the material, and the sorption results in the literature review.  Authors should find a gap for their research and explain the aim of this study with emphasizing the novelty. Well-defined aim (based on the current state-of-the-art) and novelty is missing in the article.

(3) Answer. The whole introduction was rewritten following the correct indications of the reviewer.

 (4)The research consider the application of purified and non-purified carbon nanotubes. No procedures considering the purification are presented.

(4) Answer.The purification procedure was added (Lines 122-126).

(5)Authors present N2 adsorption isotherms of both materials without any comment on it.

(5) Answer. The following comment has been added: “Adsorption isotherms show a typical profile of porous materials, but no particular differences are appreciated for purified and non-purified nanotubes. These curves are typical of physisorption type II in the classification of Brunauer - Deming - Teller. As the initial slope of the curve is rather high, this reveals the exothermicity of the N2 adsorption in the nanotubes inner channels. The first part tends to a saturation, while the sudden increase is significant of the formation of multimolecular layers.” (Lines 141-146)

(6)The paragraph „Instruments” constitutes just a list of used apparatus. Authors should give more details about the procedures used for the materials characterization using this instruments and justify their use. Besides, some characteristic is presented in the previous section, which is not consistent.

(6) Answer.The "Instruments" paragraph has been rewritten more completely (Lines 163-171).

(7)Based on the obtained adsorption data, the adsorption capacities of carbon nanotubes should be determined, not only the percentage removal. This will allow to compare the CNT with other materials used in dyes removal. Adsorption isotherms and kinetic studies should be also included. I suggest to provide more experimental points in the first 30 min of adsorption. This will allow to analyse kinetics of the sorption since the process of dye adsorption takes place mainly within the first 30 min. Authors did not provide a discussion of the obtained results in relation to results obtained by other researchers. Please improve the discussion.

(7) Answer. Figure 8 has been included in the manuscript, which shows the adsorption capacities for the various systems studied. Unfortunately, we are not in a position to add other experimental points, given the current health situation. It is our intention, however, to continue through a new work to deepen exclusively the aspect relating to the adsorption isotherms of these systems. A comment has been included in the manuscript: “For the three solutions, absorption capacity values of around 4.5, 12 and 15 mg g-1 are reached for the solutions at concentrations of 15, 37 and 45 mg/L respectively, in correspondence with the abatement close to 100%. Although, as can be guessed, the saturation of carbon nanotubes has not yet been reached in the systems analyzed, these absorption capacity values are already sufficiently important when compared with other materials used such as iron oxide nanoparticles (IONPs) [91], pumice and walnut wood activated carbon [92].”(Lines 212-222)

(8)Please correct the caption of Figure 6. Figure 8 is not consistent. Please use the same colour for one material type, and the same order of material appearance in the figure.

(8) Answer.The caption of figure 6 has been corrected and the figure has been modified.

(9)In line 159 authors write that here is a linear increase in speed which increases as the concentration of the dye solution increases. However the figure 10 does not proove it.

(9) Answer. The sentence has been made clearer.

(10)The usage of TG/DTA method in order to characterize carbon nanotubes after dye adsorption is questionable because it does not provide valuable information. What was the point of doing this study?

(10) Answer.  For a better explanation the following sentence has been added to the manuscript:” The study of the thermal characteristics of the samples was carried out, to confirm the adsorption of the dye and concomitantly to investigate both variations in weight loss and interactions between carbon nanotubes-dye molecules as a function of the process parameters.” (Lines 257-259).

(11)It should be explained in the materials and methods section. What was the FTIR spectra of the carbon nanotubes? It will help to analyse the results. Spectra presented in figures 15 and 16 are difficult to compare because the x axis is not consistent in them and main peaks are not marked.

(11) Answer. Reading the referee's notes, we realized that the spectra at our disposal do not make a significant contribution to the work and we therefore decided to cut the part relating to FTIR analyzes. Following his notes, we decided to cut the part relating to FTIR analyzes. We realized that the spectra at our disposal do not make a significant contribution to the work.

(12)What is the added value of determining different kind of carbon in the liquid samples (TOC)? No discussion of the obtained results is provided. The same question considers the results presented in the figure 17. Maybe it is not clear for the reader because the article has not well-defined aim of the study.

What was the aim of doing this research set?

(12) Answer. For a better explanation the following sentences had been added to the manuscript: “The determination of the organic and inorganic carbon content inside the solutions after the adsorption process was carried out to verify any degradation phenomena of the dye.”(Lines 309-310). “A further study was carried out to verify the reversibility of the process and to seek further evidence of degradation phenomena of the dye.” (Lines 322-323).

(13)Please limit self-citations, in the whole text more than 30% are self-citations.

(13) Answer.The bibliography has been supplemented with new references.

Reviewer 2 Report

In this manuscript, the authors demonstrated the purification of reactive black-5 dye-contained water by carbon nanotubes. Multi-techniques were utilized for characterizing the solid and liquid phases before and after the adsorption tests. It was found that the synthesized carbon nanotubes exhibited high adsorbing capacity for the removal of the reactive black-5 dye from water. It is a complete work although the novelty and significance have relative limitations. The experiments are good-designed and obtained results are convincible. All the conclusions are supported by the presented data. Therefore, this manuscript is recommended for publication at Materials after major revisions.

Special comments for the revisions:

  1. The title could be changed to “Treatment of reactive black-5 dye-contaminated water with carbon nanotubes”.
  2. The part of “Introduction” should be improved greatly. More information on the applications of carbon-based nanomaterials for water treatment should be introduced and discussed. In addition, the novelty and significance of this work should be more clear.
  3. It is better if the authors could provide a scheme to indicate the key idea of this work in the experimental section.
  4. The synthesized carbon nanotubes should be characterized with TEM or AFM techniques.
  5. Figure 3 and 4 could be combined together to one figure. Usually, in the experimental section, it is not necessary to present some experimental results.
  6. In Figure 9, it is suggested for the authors to add error bars for all the histograms.
  7. Figure 15 and 16 could be combined together.
  8. How to evaluate the treatment performances of carbon nanotubes? The authors are suggested to add more comparison with other materials to show the advantages of carbon nanotubes for treating black-5 dye.
  9. There are too may typos in the text. The English of this manuscript should be improved greatly.

Author Response

Answer to referee 1

Dear Reviewer,

thank you for your time spent reviewing our manuscript and for giving us important tips for improving it. The manuscript has been revised in many parts. The changes have been highlighted in yellow. Here are the answers to your suggestions point by point. We hope that in this form the manuscript will find your approval to be considered for its publication.

We thank you and send you our best regards.

Answers

(1)The title could be changed to “Treatment of reactive black-5 dye-contaminated water with carbon nanotubes”.

(1) Answer.  The title has been changed

(2)The part of “Introduction” should be improved greatly. More information on the applications of carbon-based nanomaterials for water treatment should be introduced and discussed. In addition, the novelty and significance of this work should be more clear.

(2) Answer. The whole introduction was rewritten following the correct indications of the reviewer.

(3)It is better if the authors could provide a scheme to indicate the key idea of this work in the experimental section.

(3) Answer. The following schematic sentence was added in the manuscript: “The experimental activity consisted of several consecutive phases: (a) procurement, preparation and preliminary characterization of raw materials; (b) execution of adsorption tests with carbon nanotubes in solutions with different concentration of dye, for different contact times and different stirring speeds; (c) characterization of the phases after the adsorption process” (Lines 105-108)

(4)The synthesized carbon nanotubes should be characterized with TEM or AFM techniques.

(4) Answer. The nanotubes used have been previously characterized in our other articles and references have been inserted. The following sentence was reported in the manuscript: “The multiwall carbon nanotubes used in this study are the same ones that have been synthesized and characterized in our previous publications; for further information, please refer to these articles reported in the references [61, 83].” (Lines 111-113)

(5)Figure 3 and 4 could be combined together to one figure. Usually, in the experimental section, it is not necessary to present some experimental results.

(5) Answer. Figures 3 and 4 have been combined.

(6)In Figure 9, it is suggested for the authors to add error bars for all the histograms.

(6)Answer. In Figure 9  error bars have been added.

(7)Figure 15 and 16 could be combined together.

(7) Answer. Figures 15 and 16 have been cut out following a revision of the manuscript.

(8)How to evaluate the treatment performances of carbon nanotubes? The authors are suggested to add more comparison with other materials to show the advantages of carbon nanotubes for treating black-5 dye.

(8) Answer. The following sentence has been added in the text:”In this work, the removal of the azo dye called Reactive Black-5 was studied, using carbon nanotubes. The use of carbon nanotubes as adsorbent materials has the advantage of generally being fast and cheap, as they can be easily regenerated and reused. Furthermore, they allow not to introduce, in the systems to be purified, neither new substances, which can subsequently be harmful to the environment, nor substances that can react with other molecules [88-90].”(Lines 92-97).

Also, the following sentence is present in the manuscript: “Another important aspect to underline is the possible recycling of post-treatment nanotubes: as seen from the thermal analyses, the combustion temperature of the dye is lower than that of the nanotubes, so exposing the post-treatment nanotubes to temperatures of around 500 °C it is possible to destroy the adsorbed molecules without damaging the nanotubes, which can be subsequently reused (Lines 347-351).

(9)There are too may typos in the text. The English of this manuscript should be improved greatly.

(9)The manuscript will be submitted to MDPI's English editing service.

Round 2

Reviewer 1 Report

I would like to thank the Authors for their efforts made in order to improve their manuscript. Tables should be formatted according to the journal standard

Author Response

The manuscript has been revised.

Best regards

Reviewer 2 Report

The authors made great improvement according to the comments and suggestions of the referee. I am satisfied with these changes and therefore recommend its acceptance with the current version.

Author Response

Best regards